# Ultrashort Implants, Alternative Prosthetic Rehabilitation in Mandibular Atrophies in Fragile Subjects: A Retrospective Study

**DOI:** 10.3390/healthcare9020175

**Published:** 2021-02-06

**Authors:** Giovanni Falisi, Carlo Di Paolo, Claudio Rastelli, Carlo Franceschini, Sofia Rastelli, Roberto Gatto, Gianluca Botticelli

**Affiliations:** 1Department of Life Health and Environmental Sciences, University of L’Aquila, 67100 L’Aquila, Italy; giovanni.falisi@univaq.it (G.F.); claudio.rastelli@univaq.it (C.R.); sofiarastelli3@gmail.com (S.R.); roberto.gatto@univaq.it (R.G.); 2Department of Oral and Maxillo-Facial Sciences, “Sapienza” University of Rome, 00185 Rome, Italy; carlo.dipaolo@uniroma1.it (C.D.P.); franceschinicarlo93@gmail.com (C.F.)

**Keywords:** ultrashort implants, immediate loading, full-arch rehabilitation, neutral zone

## Abstract

This study aimed to evaluate the effectiveness of using ultrashort implants in the rehabilitation of jaws of fragile patients. The aim of the study was to retrospectively evaluate the survival rate of full-arch prosthetic rehabilitation on ultrashort implants, length 4 mm, 4 mm in diameter in the premolar and canine area and 4.5 mm in diameter in the molar area, with the insertion torque of 60 Nw and immediate loading. Nineteen patients were evaluated for 3 years clinically and radiographically. The significant majority of the patients at the 3 year follow-up (T4) presented a stable and functional implant-supported prothesis, and the survival rate of the implants was 85%, with a loss of 16 implants on 114 implants. The combination of the innovative implant surfaces and the correct project of the prostheses, with the related implant connection, determined a different timing in the therapy, allowing to obtain an immediate loading, which is currently demanded by patients. This and recent reports on short and ultrashort implant usage in atrophic jaws offer a good solution in critical cases. In conclusion, within the limits of the study, the full-arch rehabilitation with immediate loading on ultrashort implants showed good results with few postoperative complications and related low biological cost.

## 1. Introduction

The rehabilitation of edentulism in atrophic jaws represents a difficult situation to face, especially in fragile patients [1]. Indeed, the prosthetic treatment most required by patients is the implant-supported one [1].

The implant-supported prosthetic solutions require a bone quantity able to host the implant fixture and, in cases of atrophic jaws, ridge augmentation is required to place a fixture of “normal” length (≤10 mm) [2]. These kinds of interventions are not risk-free, and the placement of shorter lengths represents a valid alternative to rehabilitate atrophic jaws [3].

The definition of a short implant is still a topic of debate; some researchers define “short implants” as fixtures having a length ranging between 7 and 10 mm, whereas others define them as fixtures presenting an intrabony length of 8 mm [4].

Indeed, some researchers consider 10 mm or less as short, while others propose <8, <7, or 6 mm as truly short [5,6]. Nevertheless, other researchers agree that ultrashort implants are 4 mm long [7,8,9].

Due to the biological and economic advantages in using this type of fixture to prosthetically rehabilitate an atrophic jaw, several studies reported at various follow-up points the survival rates of fixed prostheses implant supported, showing their efficacies [10,11,12].

Indeed, the consensus report of the Group 1 ITI (International team for Implantology) in 2018 evaluated the survival rates of short implants as similar to longer implants, with a lower rate of post-intervention complications if compared to the longer implants placed together with the procedure of bone grafting [13].

The success of the rehabilitation of an edentulism relies on the primary stability of the implant and on the correct project of the prosthesis, which should follow the masticatory dynamic of the patient, especially in full-arch rehabilitation [14].

In the Group 1 ITI consensus report, it is indeed stated the survival rate of short implants is influenced by the functionality.

Therefore, the construction of overdentures should be done with respect to the physiological state of the patient, known as the neutral zone, so that harmful masticatory forces do not affect the implants [15].

The combination of innovative implant surfaces and the correct project of the prostheses, with the related implant connection, determined a different timing in the therapy, allowing to obtain an immediate loading [16,17,18].

Immediate loading indeed relies on many factors, such as the number of implants and the biological bone response (mostly the primary stability) [19,20].

In case of short implants, the improvement of the surfaces allows an optimal primary stability and, as a consequence, the possibility of immediate loading.

The aim of the study was to retrospectively evaluate at different timings the survival rate of full-arch prosthetic rehabilitation on ultrashort implants (length 4 mm), with immediate loading.

## 2. Materials and Methods

### 2.1. Patient Characteristics

Between 2015 and 1027, approximately 150 patients that needed a total implant-supported inferior rehabilitation were examined at the Department of Odontostomatology of the University of L’Aquila. In order to achieve a successful prosthetic rehabilitation, all patients were studied through our new clinical protocol, which included a first gnathological evaluation through RC-TMD (Research Criteria- Temporomandibular Disorders) to evaluate the stomatognathic functional status [15].

The following exclusion criteria were applied for the selection of the study group: patients smoking more than 12 cigarettes per day; patients with high risk factors; patients on bisphosphonate therapy; patients who underwent radiotherapy of the head and neck region in the previous 12 months; patients with temporomandibular and/or parafunctional dysfunctions; pregnancy; poor oral hygiene or inability to undergo the follow-up protocol. The following inclusion criteria were applied for the selection of the study group: patients over the age of 18; lower total edentulous patients or patients made edentulous due to the presence of severely compromised teeth; mandibular atrophy with residual crest not less than 5 mm from the roof of the mandibular canal; authorization by the patient to participate in the study.

### 2.2. Follow-Up Protocol

Of the 150 patients, 40 were excluded because they were severe smokers, 12 were on bisphosphonate therapy, 41 had high risk factors, five had temporomandibular osteoarthritis, 25 were unable to follow the protocol, and eight had poor oral hygiene. The resulting sample consisted of 19 subjects, of which 11 were men aged between 62 and 77 with an average age of 69.5 and eight were women aged between 61 and 71 with an average age of 66.

According to the Italian legal system no. 127 of 1996, these patients are counted as fragile subjects, not only due to their age, but also due to their economic status and local biological conditions. All selected patients were completely edentulous or to be rendered edentulous as the residual elements could not be used as a source of anchorage.

The sample was subjected to preventive anamnestic tests and then underwent imaging tests for therapeutic planning.

The Ethics Committee approved the study (n. 55/2018.19), and all patients signed an informed consent form.

The sample underwent implant surgery with the use of ultrashort implants (4 mm long, 4 mm in diameter in the premolar and canine area and 4.5 mm in diameter in the molar area, with the insertion torque of 60 Nw) Twinkon4, TEKKA, Global D. This type of implant is a grade 5 titanium alloy (TiAI6V4), sandblasted and double-etched, with a surface roughness of 1–2 μm.

A temporary prosthesis with immediate loading was applied to all patients, since the characteristics of the implant design and of the transmucosal type allowed no solution other than to solder the implants, as already experimented in other implant prosthetic protocols.

### 2.3. Implant Characteristics and Surgical Protocol

The implant surgery involved the use of the ultrashort implants (4 mm long) Twinkon4, TEKKA, Global D. This type of implant is a grade 5 titanium alloy (TiAI6V4), sandblasted and double-etched, with a surface roughness of 1–2 μm.

All the considered patients were fitted with a temporary prosthesis with immediate loading.

The surgical protocol involved oral administration of antibiotic therapy with 2 g of amoxicillin 1 h before and then 1 g every 12 h for 5 days.

Before starting the surgery, the perioral surface was disinfected with povidone iodine (10% Betadine), while the patients were intraorally rinsed with a 0.2% chlorhexidine solution for 60 s. The loco-regional anesthesia was performed with 4% articaine with epinephrine 1:100,000 (CITOCARTIN “100” Molteni Dental).

The protocol included the use of a positional guide for the implant placement.

The positional guide was applied, and a punch was made on the crestal mucosa to find the positioning of the implant; then, a full thickness crestal incision was made to skeletonize the underlying bone (Figure 1).

The Tekka protocol was used for the positioning of the ultrashort implants (TwinKon4 Tekka Global D). This protocol requires the use of drill bits with increasing diameter up to the diameter necessary for the insertion of the corresponding implant. A total of 38 TwinKon 4 implants, with a 4.5 mm diameter, were placed in the area 3.6 and 4.6. A total of 76 TwinKon 4 implants, with a 4 mm diameter, were placed in the area of the premolar and canine. The insertion torque was 60 N.

Once the mucosa was sutured with detached or X-shaped stitches, the temporary prosthesis was fixed on the temporary abutment, taking care to make the peri-implant area easy to clean using routine oral hygiene procedures (Figure 2).

### 2.4. Prostheses Project

The operating protocol first envisaged the survey of a conventional analogue impression in alginate (Kromopan Lascod s.p.a) of the dental arches; then, an articulation base was built for the detection of the neutral space and for the vertical dimension. This method used TENS (Trans Cutaneous Electrical Nerve Stimulation) stimulation (J5 Myomonitor^®^ TENS Unit device of Myotronics-Noromed, Inc., Tukwila, WA, USA) with electrodes (Myotrode SG Electrodes^®^, Myotronics-Noromed, Inc., Tukwila, WA, USA), whereas resin was used for the detections (Sapphire Resin, Myoprint) (Figure 3).

Subsequently, both a conventional prosthesis and a baryta resin duplicate were made to perform the cone beam.

In order to find the implant position, the DICOM (Digital Imaging and Comunications in Medicine) images were processed with the acquisition program (3 shape Implant Studio) to create a positional template for the surgical phase.

The implant position was guided by a functional gnathological evaluation and by a series of prosthetic reference factors such as the distribution of the occlusal load and the neutral space (which is the space where the resultant of the strength of the muscles, tongue, and cheeks is equal to zero).

### 2.5. Follow-Up

Seven days after surgery, a clinical examination of the patient was performed, in which the healing status of the mucosa and the implant stability were analyzed and the sutures were removed. After 4 months, a cone beam was performed to assess whether the bone healing status had no problems (Figure 4b), before proceeding with the realization of the final prosthesis with a screwed bar. An OPG (Orthopantomography) was carried out 1 year (Figure 4c) and an X-ray OPG was carried out 3 years (Figure 4d) after the realization of the definitive prosthesis.

## 3. Statistical Analysis

The primary outcome was the number of patients who, at the 3 year follow-up, had a stable prosthesis supported by at least four implants, evaluated using chi-square test. The secondary outcome was the implant survival at T1, T2, and T3, statistically evaluated using the Fisher exact test, and the endpoint was the survival rate of the implants at T4. Any statistically differences were considered significant at a *p*-value <0.05.

## 4. Results

Nineteen patients were included in the retrospective analysis, 11 males and eight females, with an age ranging from 64 to 77. Implants were mostly lost 1 week after loading and at the fourth month follow-up.

### 4.1. Primary Outcome Results

Eighteen patients at T4 presented a stable prothesis supported by ≥4 implants. As shown in Table 1, the chi-square test showed a statistically significant difference (*p* < 0.05).

### 4.2. Secondary Outcome Results

The Fisher exact test applied to determine the implant loss occurrence at the considered follow-up periods (T1, T2, T3) showed a statistically significant difference (Table 2).

### 4.3. Endpoint

At T4, the survival rate of the implants was 85%, with a loss of 16 implants from a total of 114 implants.

## 5. Discussion

The retrospective follow-up of patients who underwent full-arch rehabilitation with ultrashort implants showed how the critical points of rehabilitation were the first week and the 4 months after implant placement and the related prosthetic loading, as observed in the literature [4].

Overall, the unsuccess of the rehabilitation was not frequent, and basically only one patient could not rely on this type of prosthetic rehabilitation.

If the surgical intervention on jawbones is considered as a minor type on surgery and with a very low risk, the diagnosis and planning stages are fundamental for rehabilitation, which can be comfortable and satisfying for the patients [16].

The oral cavity, with its sensitivity and peculiar functionality due not only to its occlusion and masticatory function but also to its phonatory, respiratory, and tasting function, requires a careful rehabilitation plan [18].

Currently, the rehabilitation of small edentulism requires an established plan, facilitated by the presence of other dental elements. Edentulism cases requiring a complete denture rehabilitation are challenging for both anatomical and functional reasons [21]. Indeed, the bone availability is conditioned by the degree of alveolar resorption, which places the noble structures such as the inferior alveolar nerve and the maxillary sinus at risk [22]. In addition, risks of fracture are higher in cases of severely atrophic jaws during surgical intervention [21].

The loss of the vertical dimension modifies the functionality of the masticatory organ, and spotting the neutral zone might be difficult [16].

Several surgical techniques are available to improve the volumetric bone availability. Vertical ridge augmentation, large sinus lift, xenograft interventions, and placing of pterygoid implants or zygomatic implants are some of the most famous and new techniques proposed in the literature [2]. However, the high cost and the morbidity risks associated with these techniques are not appealing from both the clinician’s and the patient’s point of view [10,13,23,24].

The improvement of implant surfaces and the establishment of defined surgical protocols allowed short implants to be a reliable alternative to rehabilitate small edentulism in atrophic jaws; however, the rehabilitation of full edentulism using full-arch prostheses has also been reported with a certain degree of success.

Overall, the primary outcome showed the success of the ultrashort implant-supported prosthetic protocol. However, the rate of implant loss was slightly higher, if compared with the survival rate reported by the Group 1 ITI Consensus Report, which reported a survival rate ranging from 86.7% to 100% [13]. However, the reported data included two different variables which were not considered or studied in the literature: the loading time (immediate) and the type of prothesis (full-arch rehabilitation).

The available studies in the literature reported the survival rate of short and ultrashort implants with immediate or delayed loading supporting single crowns [23,24,25,26,27].

On the other hand, few case reports are available in the literature reporting full-arch prothesis supported by ultrashort implants and placed with immediate loading [17,19,28].

Indeed, Falisi et al. reported a full-arch rehabilitation on ultrashort implants with no postoperative complication and success of the prosthesis integration, in functional and aesthetic terms. In this case, the prosthesis was planned according the neutral zone of the patient [23].

Pistilli et al. also reported the case of a rehabilitation of a severe atrophic mandible on implants no longer than 4 mm, highlighting how this therapeutical choice reduced timings, costs, and postoperative complications. In this case, the occlusal loading was the most balanced [21].

However, the data of a previous study showed how immediate loading can affect the survival rate of implants but not the final prothesis success, showing how the correct distribution of the occlusal forces in the period after implant insertion can positively influence the final outcome [16].

Indeed, the healing period after the surgical stage is the most crucial one, since the bone tissue is submitted to a remodeling process balanced between the resorption and production of the mineralized tissue [25].

The theme of immediate loading on short implants is still under investigation among scientists and clinicians. Indeed, Weerapong et al., on the basis of the results of their randomized clinical trial, stated that immediate loading on short implants gives survival rates comparable to immediate loading on longer implants, together with the use of new technologies and CAD/CAM (Computer Aided Design/Computer Aided Manufacturing) replanning of the prosthetic device [26].

Within the limits of the current study, regarding the sample size and the observational nature of the study, the full-arch rehabilitation with immediate loading on ultrashort implants showed good results with few postoperative complications and related low biological cost.

Future prospective studies and randomized clinical trials with a larger sample size are needed to evaluate the survival rate of full-arch rehabilitation on short and ultrashort implants with appropriate occlusal loading.

## 6. Conclusions

Results from our study suggest the adequate clinical performance of short implants after 3 years of immediate loading. Thus, 4 mm ultrashort implants with a diameter of 4 and 4.5 mm could represent a good alternative to rehabilitate edentulous atrophic jaws or jaws with residual elements that cannot be used as a source of anchorage. 

## 7. Ethics

The study was approved by the Internal Review Board of University of L’Aquila, n. 55/2018.19.

## Figures and Tables

**Figure 1 healthcare-09-00175-f001:**
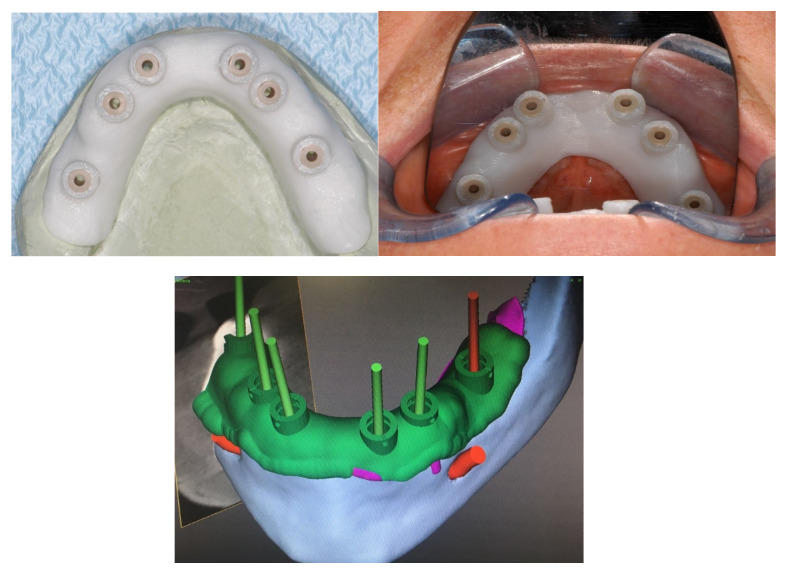
Positional guide for the implant placement.

**Figure 2 healthcare-09-00175-f002:**
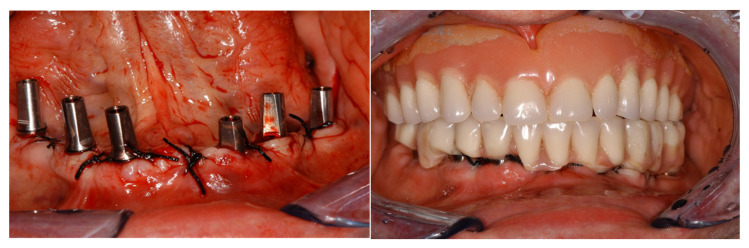
The temporary prosthesis was fixed on the temporary abutment.

**Figure 3 healthcare-09-00175-f003:**
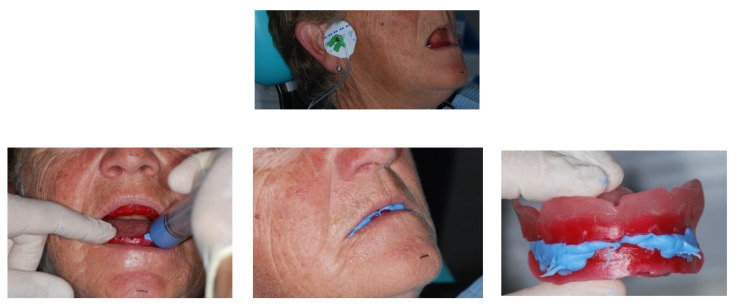
TENS (Trans Cutaneous Electrical Nerve Stimulation) stimulation with electrodes and detection with Myopront resin.

**Figure 4 healthcare-09-00175-f004:**
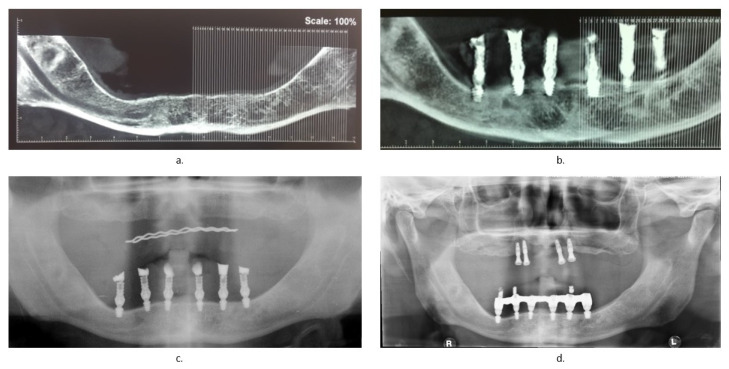
Radiographic checks: (**a**) cone beam before surgery; (**b**) cone beam 4 months after surgery; (**c**) OPG (Orthopantomography)1 year later; (**d**) OPG 3 years later.

**Table 1 healthcare-09-00175-t001:** The significant majority of the patients at the 3 year follow-up presented a stable and functional implant-supported prothesis. The chi-square test showed the significant success of the proposed protocol.

Primary Outcome: Success of the Prothesis			
Prothesis Outcome	Frequency (*N*)	Percentage (%)	*p*-Value
Successful prosthesis	18	5.26	<0.05
Unsuccessful prosthesis	1	94.73	

**Table 2 healthcare-09-00175-t002:** The Fisher test was significant with a *p*-value <0.05 regarding the time of implant loss occurrence.

Implant Loss Occurrence	T1 (1 Week)	T2 (4 Months)	T3 (12 Months)
Loss	3 (15.79%)	7 (36.84%)	0 (0)
No Loss	16 (84.21%)	12 (83.16%)	19 (100%)

## Data Availability

Data will be availble upon resonable request to the corresponding author.

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
