# Peer review of "Ultrashort Implants, Alternative Prosthetic Rehabilitation in Mandibular Atrophies in Fragile Subjects: A Retrospective Study"

_healthcare, 2021, doi:10.3390/healthcare9020175_

Round 1

Reviewer 1 Report

Dear Authors,

I would like to thank You for the efforts placed to conduct this study. The use of ultra-short implants to rehabilitate atrophic jaws is an interesting topic of research, however I evidenced some critical issues.

First of all, this is not a case-report as you described the article.

I don’t understand why did You load immediately ultra-short implants in a full-arch rehabilitation in edentulous patients: this was not an evidence-based protocol.

INTRODUCTION

  • Lines 26-28: please check the English.
  • Lines 29-33: please check tenses (You used past and present in the same phrase).

MATERIALS AND METHODS

  • In the title you said “fragile patients”, please explain in this section what characteristics You considered to classify a subject as fragile-patient.
  • A section in which you explain inclusion/exclusion criteria lacks.
  • In the statistical analysis You should also describe the primary outcome- survival rate and related criteria to define it.
  • Why did You use z-test to compare the number of patients?
  • Why did You use Poisson regression? Which was your dependent variable and independent one?
  • In the reported OPG images I couldn’t find any atrophic ridge, the case presented showed a good availability of bone in the anterior mandible. It seems enough to position standard implants.

RESULTS

  • You did not report survival rate.
  • Half of included patient lost implants, and it was not comparable to survival rate of short implants presented in literature, which was higher.
  • It was not clear the outcomes of your regression analysis.
  • Tables are unclear; more details are needed to understand and interpret your results.

DISCUSSION

  • English must be checked and corrected
  • Results of your research were not discussed.
  • Half of included patients lost implants and this is not a good results as You said in the discussion.

Author Response

my team and I have reviewed the scientific work according to your indications.

Reviewer 2 Report

It is a retrospective study on the survival of immediate mandibular restorations on 6 ultra-short implants (length 4 mm), the follow-up is three years.

The study is interesting, although the authors should clarify some aspects of the method and above all provide the results transparently: More than half patients lose an implant, 14% of the implants were lost, one of the 19 patients could not be rehabilitated with a prosthesis.

Authors conclude that: “As reported in table 1 and in table 2, the difference between the patients who lost implants and the difference between the numbers of implants survived at the considered follow up were not statistically significant (P > 0.01)”. This may lead to confusion.

Abstract

The inclusion of truly relevant information can improve the Abstract:

  • What are the different timings?
  • Specify the number and diameters of the implants used (6 implants 4mm long are used in each, with a diameter of 4.5mm or 4mm placed at 60Nw.)
  • The results should be clearer: 19 patients were included, 10 lost at least 1 implant. 114 implants were placed, 16 were lost representing 14% of the total. During the first week 3 implants were lost and month 4 a further 13 implants. One patient lost 5 implants and could not perform the prosthesis
  • I don’t think this paragraph should be included in the abstract “The combination of the innovative implant surfaces and the correct project of the prostheses, with the related implant connection, determined a different timing in the therapy, allowing to obtain an immediate loading, which is nowadays demanded by patients. This and recent reports on short and ultra-short implants usage in atrophic jaws offer a good solution in critical cases.”
  • In conclusion within the limits of the study, the full-arch rehabilitation with immediate loading on ultrashort implants showed good results with low post-operative complications.
  • I would not include references to the economic cost that is not covered in the article

Introduction

 ‘normal’  length (<10 mm). Check I think is a mistake

Materials and Methods

How do you diagnose parafunctions, in an edentolous patient? explain

What are the different timings? explain

Are/were they all evaluated at 3 years? confirm

Always in all cases, implants could be placed at 60Nw? confirm

I think the results of the data analysis should be better explained likewise the statistical analysis reviewed and interpreted.

Author Response

My team and I have reviewed the scientific work according to your indications.

Reviewer 3 Report

Manuscript ID: healthcare-1068306

Type of manuscript: Case Report

Title: Ultra-short implants, alternative prosthetic rehabilitation in mandibular atrophies in fragile subjects: a retrospective study

This manuscript was intended to addresses a very up-to-date topic occurring now in implantology and prosthetics. Unfortunately, the manuscript suffers from many shortcomings and does not meet the standard required for publication consideration in the scientific journal.

The main concerns are as follows:

  • The abstract is vague and does not include the basic information about the results coming out of this study – e.g. how many implants were lost, rate survival etc.
  • OPGs presented in Figure 4 do not belong to the patient described in inclusions criteria in the M&M section: “mandibular atrophy with a residual ridge not less than 4 mm from the mandibular canal.” In fact, the long distance can be seen between the intraosseous ends of 4mm long implants and mandibular canal! So this patient does not have mandibular atrophy according to the mentioned definition.
  • It is unclear why the surgeon did not implant the longer implants (e.g. 8-10mm) above the mandibular canal and much longer (11-14mm) in the anterior region of a mandible (devoid of mandibular canal).
  • In the M&M section the description regarding the patients selection for the study group is unclear and should be reformulated and developed – “150 patients, needing a lower total supported implant rehabilitation, were examined at the Dental clinic of the University of L'Aquila, between 2015 and 2017. Among all the patients, 19 benefited from the clinical protocol for the designing and construction of the prosthesis according to their neuromuscular scheme.” What about other patients? - does it mean that only 19 patients underwent this protocol?
  • The table 1 and 2 are unclear and should be described more closely. The sentence - ” Z test to compare the number of patients who lost implants.” - does not explain much.
  • Similar with the last sentence of the Results section – “the difference between the patients who lost implants and the difference between the number of implants survived at the considered follow up were not statistically significant (P > 0.01).” Does it suppose that the survival rate was successful?
  • According to Group 1 ITI Consensus Report – a reference mentioned in this manuscript [Jung RE, Al-Nawas B, Araujo M, Avila-Ortiz G, Barter S, Brodala N, Chappuis V, Chen B, De Souza A, Almeida RF, Fickl S, Finelle G, Ganeles J, Gholami H, Hammerle C, Jensen S, Jokstad A, Katsuyama H, Kleinheinz J, Kunavisarut C, Mardas N, Monje A, Papaspyridakos P, Payer M, Schiegnitz E, Smeets R, Stefanini M, Ten Bruggenkate C, Vazouras K, Weber HP, Weingart D, Windisch P. Group 1 ITI Consensus Report: The influence of implant length and design and medications on clinical and patient-reported outcomes. Clin Oral Implants Res. 2018 Oct;29 Suppl 16:69-77. doi: 10.1111/clr.13342. PMID: 30328189.]. Short implants (≤6 mm) revealed a survival rate ranging from 86.7% to 100%, whereas standard implant survival rate ranged from 95% to 100% with a follow-up from 1 to 5 years. In this study survival rate was 85,96% after 3 years. This number should be mentioned in the Results section and discussed in the Discussion section.
  • Apart from the conclusion’s part the Discussion section is almost devoid of any references to the results obtained by this study.
  • The conclusion about “economical costs” do not fully arise from the results of this study because the study mentions nothing about any costs, not to mention cost-efficiency analysis based on the presented material.

Author Response

(The authors gave the same response as above.)

Reviewer 4 Report

The authors present a retrospective comparative study.  The study aimed to evaluate the effectiveness of using ultra short implants in the rehabilitation of atrophic jaws of fragile patients. The manuscript by Falisi et al, titled " Ultra-short implants, alternative prosthetic rehabilitation in mandibular atrophies in fragile subjects: a retrospective study", the manuscript is well written, but for the publication there are some points to be added/changed. The manuscript has no conclusions to correlate with the objective. However, the images presented in the manuscript are from a single clinical case and decontextualized:

  • In line 90-92: “The positional guide was applied and a punch was made on the crestal mucosa to 90 find the positioning of the implant, then was made a full thickness crestal incision to skel-91 etonize the underlying bone (Figure 1).”

The image presented in the manuscript is an image with abutments.

  • In line 102-104: “Once the mucosa was sutured with detached or X-shaped stitches, the temporary prosthesis was fixed on the temporary abutment, taking care to make the peri-implant area easy to be clean by the routine oral hygiene procedures (figure 2).”

The image shows a badly adapted temporary prosthesis with poor finish which implies an increase in dental plaque.

  • In line 110-114: “This method used TENS stimulation (J5 Myomonitor® TENS Unit device of Myotronics-Noromed, Inc., Tukwila, WA, USA) with electrodes (Myotrode SG Electrodes®, Myotronics-Noromed, Inc., Tukwila, WA, USA) and for the detections was used resin (Sapphire Resin, Myoprint) (Figure 3).”

TENS stimulation is not represented in any of the images 3.

Some points to be added/changed as following:

Introduction:

Add more rationals for conducting this study and why you used the selected implant systems.

what is the clinical relevance by conducting this study?

Materials and Methods

The images must have a better match or must be removed from the manuscript. Include a justification for the use of the presented method in the described case, and not using larger implants due to the mandibular bone availability verified in the X-ray.

Statistical analysis

There is a lack of a more consistent and detailed statistical analysis, since it is a retrospective study.

Discussion

The authors should compare this technique with the most common techniques used in atrophic jaws, referring advantages and disadvantages.

Note: This manuscript should be transformed into a Clinical Case Report article.

Author Response

(The authors gave the same response as above.)

Round 2

Reviewer 2 Report

In my opinion the manuscript has been significantly improved

Author Response

Dear reviewer thank you for the comments.

Best Regards 

Dott. G. Botticelli

Reviewer 3 Report

Manuscript ID: healthcare-1068306

Type of manuscript: Short communication

Title: Ultra-short implants, alternative prosthetic rehabilitation in mandibular atrophies in fragile subjects: a retrospective study

The manuscript has been revised, but still requires more improvements before  publication consideration as scientific article.

The main concerns are as follows:

  • The abstract – no concerns now.
  • The introduction – has to state precisely the definitions of “fragile patient”, “short implant” and “ultrashort implant” with appropriate references. The given definitions in the sentence: “The definition of short implant is still a topic of debate: some researchers define as “short implants” fixtures having a length ranging between 7 and 10 mm, and others those fixtures presenting an intrabony length of 8 mm (3).” Are insufficient and has to be developed.  8mm is between 7 and 10mm, so the mentioned sentence is unclear. It seems reasonable to give more references when mentioning “and others” in regard to several authors.
  • Usually there are captions to illustrations – “Figure 1.” Is not enough.
  • Consequently in one manuscript one style should be used - for instance, under the illustration should be “Figure 1.” while in the text “Fig. 1”. Now is different - see the lines number 135, 147, 159, 181.
  • There is still a huge problem with OPG presented as Fig. 4. – The authors of this study should explain why in this case (presumably in the fragile patient) the standard length implants were contraindicated and were not used. The answer should address the medical practice point of view not the legal one. The explanation: “According to the Italian legal system no.127 of 1996 these patients are counted as fragile subjects, not only for their age, but also for their economic status and local biological conditions. All selected patients were completely edentulous or to be rendered edentulous as the residual elements could not be used as a source of anchorage.” answers why this patient was treated according to ultrashort implants protocol. Nevertheless, the medical arguments may indicate that the applied protocol was in this case inappropriate. Alternatively the authors may replace the presented OPG if not belonging to the same patient as the presented case (at the Conclusions section the sentence: “Additionally, this case will continue to be observed over time to report any changes in the results” suggests it was only a case report. In my opinion the collected material allowed for more than a case report but the authors have to be more consequent all over the manuscript) by a more appropriate one.
  • One of the limitations of this study is relatively short observation period as well.
  • In the Conclusions section “This approach could reduce operative times, possible complications, postsurgical morbidity” might be true but did not come out of your results. You did not measure the surgical time nor estimated possible complications nor evaluated the postsurgical morbidity. Hence, you are not entitled to form these conclusions on the ground of the presented results.

Author Response

My team and I thank you for the review. We have modified our scientific work following his indications.

Reviewer 4 Report

The new version of the article “Ultra-short implants, alternative prosthetic rehabilitation in mandibular atrophies in fragile subjects: a retrospective study Authors: Giovanni Falisi, Carlo Di Paolo, Claudio Rastelli, Carlo Franceschini, Sofia Rastelli, Roberto Gatto, Gianluca Botticelli”, transformed into a Short communication, is within the magazine's rules and the corrections that were requested were made. The images have been replaced and there is an improvement in the proposed text. Therefore, the article has conditions to be published.  

Author Response

(The authors gave the same response as above.)
